# Exploring the lived experience of families with a COVID-19 positive child: The journey from a critical grounded theory approach

Jessica Kaufman[1,2]*, Kathleen L. Bagot[1], Tria Williams[1], Carol Jos[1], Margie Danchin[1,2,3]

**1** Murdoch Children's Research Institute, Vaccine Uptake Group, Parkville, VIC, Australia, **2** Department of Paediatrics, The University of Melbourne, Parkville, VIC, Australia, **3** The Royal Children's Hospital, Parkville, VIC, Australia

* Jess.kaufman@mcri.edu.au

## Abstract

COVID-19 and associated public health policies have significantly disrupted the lives of both adults and children. Experiences of COVID-positive adults are well described but less is known about the experiences of families of children who receive a positive diagnosis, and the impact of public health policies on this experience. This study aimed to develop a framework to understand the lived experience of families with a child testing positive for COVID-19. We applied a qualitative study design, using grounded theory. The study took place in Melbourne, Australia between July and December 2020, during the first major Australian COVID-19 wave. Parents of children 0–18 years tested at a walk-in clinic at a paediatric tertiary referral hospital were invited to participate. Two interviewers jointly undertook in-depth interviews with parents of children who tested positive. Interviews were transcribed and two analysts used an inductive, critical realist analysis approach with NVivo and a virtual whiteboard. Results are presented incorporating a stratified reality (empirical, actual, real). Families described seven sequential stages of the COVID-19 positive testing journey: COVID-19 close to home; time to be tested; waiting for the test result; receiving the result; dealing with the diagnosis; coping with isolation; and moving forward/looking back. Our findings highlight how public health policies and messages targeting the general (adult) public were experienced by families. We provide a framework that families move through when their child tests positive for COVID-19. Within each phase, we report unmet needs and identify strategies to improve future pandemic planning for parents and children.

## Introduction

In the first two years of the COVID-19 pandemic, there have been over 520 million cases and over 6 million deaths worldwide [1]. Although adults comprise the majority of COVID-19 cases, almost one-quarter of Australian cases are in children under 19 years (23%: 834,422/3,555,970) [2]. While children generally experience asymptomatic or mild illness [3,4], COVID-19 is readily transmitted within the home [5] and to parents and others within the household.

**Data Availability Statement:** Study data may be shared on request as per the study protocol: "Beginning 12 months following analysis and article

publication, the following will be made available for use by future researchers from a recognized research institution whose proposed use of the data has been ethically reviewed and approved by an independent committee and who accept MCRI's conditions for access: Individual participant data that underlie the results reported in this article after de-identification (text, tables, figures and appendices), and study protocol, statistical analysis Plan, and PICF." Interested researchers should contact the Director of Research Ethics & Governance at The Royal Children's Hospital Melbourne on (03) 9345 5044 or Rch.Ethics@rch.org.au.

**Funding:** This work was supported by the Royal Children's Hospital Foundation. The funders had no role in study design, data collection and analysis, decision to publish, or preparation of the manuscript.

**Competing interests:** The authors have declared that no competing interests exist.

To control case numbers in 2020, public health messaging from the Australian government instructed people to get a PCR test immediately if they experienced symptoms, however mild, and then return home to wait for the result. If contacted with a positive result, people were required to participate in contact tracing to identify close contacts and isolate at home for 14 days. Isolation was defined as no interaction with others outside the house, and where possible, isolation from uninfected members of the household as well. These isolation requirements meant that any positive COVID-19 test result within a family led to significant disruption to work, education, childcare and social activities.

The overall pandemic experiences of families [6] and specific experiences of adults who test positive for COVID-19 is being investigated, but little is known about the impact of a positive test on the child, parents and families [7,8]. In one qualitative study of 11 COVID-positive adult patients in New South Wales [7], five themes were identified to describe participants' experiences in an isolation facility: knowing about COVID-19; planning for, and responding to, COVID-19; being infected; life in isolation and the room; and post-discharge life. Another qualitative study with 13 non-hospitalised adults with COVID-19 (and 1 hospitalised) divided their experiences into two broad themes: physical (i.e., physical symptoms such as fatigue, loss of taste or smell) and psychological/emotional (e.g., loss of control, fear, shame, anxiety) [8]. As new variants emerge and vaccine coverage among adults increases, COVID-19 cases in children will increase, making it increasingly important understand the needs of families.

The aim of this study was to explore and describe the experiences, information, and support needs of families with at least one child testing positive for COVID-19. We developed a framework of the sequential stages families go through when a child receives a positive COVID-19 diagnosis and conclude with recommendations to improve this experience.

## Materials and methods

### Design and context

The COVID Wellbeing Study was a mixed-methods longitudinal cohort study investigating the immediate and longer-term health and wellbeing impacts of COVID-19 on children and families tested for COVID-19 at the Royal Children's Hospital (RCH) in Melbourne, Victoria. The study recruited participants during Victoria's second wave of COVID-19 infections, between July-December 2020, from the RCH Respiratory Infection Clinic (RIC). Many general population testing centres were not initially equipped to test children under five years, so families with younger children from around Greater Melbourne attended the RIC clinic. The RIC could test all members of the family, including adults.

In this qualitative component of the larger COVID Wellbeing Study, we conducted in depth interviews with parent/s of children who tested positive for COVID-19. We used a critical realism grounded theory approach, [9] and explored how the government policies (domain of the 'real') and messaging (domain of the 'actual') about COVID-19 testing were experienced by families with a child testing positive for COVID-19 (domain of the 'empirical') [9] in Victoria, Australia.

### Participant recruitment and consent

Parents were eligible for participation in the larger COVID-19 Wellbeing study if they i) accessed the RIC for a COVID-19 test for their child/ren 0–18 years, regardless of test result; ii) provided consent to be contact about research and iii) were able to read and understand English. Families were invited to enrol via phone call and email invitation from a member of the research team. Interested participants were sent an information and consent form, along with a link to the first COVID-19 Wellbeing survey. Consent to enrol in the study was implied

via survey completion. A total of 342 children (47 COVID-19 positive, 295 COVID-19 negative) and their families were recruited into the larger study. At the time, this was the largest known cohort of COVID-19 positive children involved in a COVID-related wellbeing study in Australia.

A subsample of parents with a positive child enrolled in the COVID-19 Wellbeing study were eligible for the qualitative component of the study. Consent was sought at the end of the first survey to be contacted about participation in the family interview. A research assistant telephoned potential participants, provided a participant information statement and obtained written electronic informed consent via the REDCap eConsent function. Children testing positive and aged > 6 years were eligible to join the interview with their parent with parental consent. Age-appropriate participant information was available.

## Data collection procedure

Two researchers (JK, PhD, Public Health; TW, Paediatric Nursing, RN, M Clinical Research) jointly conducted each in-depth interview with family members, using a semi-structured interview schedule. The schedule covered experiences through the COVID-19 pandemic, having their child tested for COVID-19 and post-testing outcomes. The interview schedule was comprised of open questions with some a priori probes. Throughout the data collection period, questions and probes were refined or added, driven by interview content and initial analyses. For example, participants spoke about their experiences of contact tracing, and this was subsequently added to future interviews to explore further.

Our participant sampling was pragmatic; that is, we interviewed all eligible families with children testing positive willing to participate in interviews. As such, we were unable to undertake theoretical sampling to recruit participants to specifically inform category and/or construct development.

All interviews were conducted remotely using Zoom at a time that was convenient for participants. Before initiating audio recording, the interviewer established rapport, ensured the participant was in a comfortable and private location, and helped address any technical issues [10]. Interviews were professionally transcribed. Researcher field notes were also completed at the end of each interview, capturing interviewer reflections and early insights. At the conclusion of the interview, the family was sent a gift voucher to thank them for their time and contribution.

The study received ethics approval from the Royal Children's Hospital Human Research Ethics Committee (HREC 63636). Written informed consent was obtained prior to data collection.

## Data analysis procedure

Two researchers (JK; KB, PhD, Psychology) conducted an iterative analysis involving both inductive and deductive approaches. The core principles of grounded theory research were followed [11]. Based on interview data and field notes (i.e., concepts grounded in data), one researcher (JK, interviewer) developed the initial framework (i.e., identifying stages of the journey, initial concepts within each stage; inductive). Another researcher (KB) subsequently categorised data into codes and sub-codes within each of these time points (deductive). Interview transcripts were reviewed multiple times and comparisons were constantly made between transcripts and the interim framework [12]. Key concepts were identified and added to the coding framework. Iterative analysis included open coding, conceptual clarification, and preliminary theoretical coding. Researchers (JK, KB, TW, MD) discussed the coding and framework structure and content throughout the analysis. Initial analysis was undertaken in NVivo

(v12) [13], with preliminary results finalised using a shared, virtual whiteboard (https://ideaflip.com/). The virtual whiteboard supported adding notes, comments, and questions to consider when reviewing transcripts. Grammar-corrected quotes illustrating key experiences within each timepoint are presented. Results are presented in accordance with the consolidated criteria for reporting qualitative research (COREQ) [14].

### Researchers' reflexivity statement

Our multidisciplinary research team comprised members with medical training, clinical and social science research expertise and experience. The interviewers and analysts were from the disciplines of communications, clinical nursing, and psychology. The study PI (MD; MBBS, FRACP, PhD) brought expertise as a paediatrician and social scientist. These varying perspectives informed our discussions throughout, and the resultant findings. Interviewers were closely involved in analysis.

## Results

Of the 47 families enrolled in the COVID Wellbeing Study with a COVID-positive child, a total of 15 agreed to participate. Interviews were scheduled to be convenient for the family, and were conducted between October 2020 and July 2021, each lasting an average of 51 minutes.

### Participating families

Of 15 families participating, the mother was the sole participant in 10 interviews, and both mother and father participated in 5 interviews (Table 1). No interviews were conducted one-on-one with the child, however two children responded to a few questions from the interviewer and/or their parent during interviews. These limited data are not included in Results.

While the primary focus was a specific child who had tested positive, experiences extended throughout the family, including testing, symptoms, illness, and isolation. For some families, the only member testing positive was a child with mild symptoms, while for other families, all household family members, including other children, also became unwell.

### The COVID-positive child journey framework

The resultant conceptual framework comprises seven stages that participating families went through when their child tested positive for COVID-19 (Fig 1). Within each stage, there were key, specific experiences identified, varying between and within families. The relationship between the domains of the real, actual, and empirical is illustrated in Fig 2.

**COVID-19 close to home.** Before encountering COVID-19 themselves, participants varied in their approach to the pandemic. Some viewed it as high-risk and strictly adhered to public health measures, while others were more flexible in their adherence. In addition to the pandemic, many participants described significant life events involving health, employment, and relationships.

The initiating events that turned the pandemic into a personal reality were grouped into high and low alarm events. High alarm events were those where the child was actively exhibiting symptoms or where the parents had a high expectation of a positive test result, for example where one parent was an aged care worker at a facility with positive cases. Alternatively, low alarm events where participants had a low expectation of illness included testing at the end of a close contact quarantine period or having a child with minor cold symptoms. Deciding to get tested was immediate for some: "I was like, right, let's get to the Children's Hospital. Let's get

**Table 1. Demographic details of participating families.**

| Participant ID | Participant relationship to child | Age of COVID-19 positive child | Family structure and COVID-19 status (positive +, negative -) | Isolation period |
|---|---|---|---|---|
| 094 | Mother | 15 months | Mum (-), Dad (-), 1x child (+) | 4 weeks |
| 261 | Mother Father | 8 months | Mum (+), Dad (+), 1x child (+) | 19 days |
| 286 | Mother | Under 5 years | Mum (-), Dad (-), 1x child (+) | 4 weeks |
| 214 | Mother | 11 months | Mum (-), Dad (-), 1x child (+) | 24 days |
| 230 | Mother | 3 years | Mum (+), Dad (+), 2x child (+) | 3 weeks |
| 211 | Mother | 4 years | Mum (-), Dad (-), 1x child (+) | 4 weeks |
| 204 | Mother | 12 years | Mum (-), Dad (-), 1x child (+) | 3.5 weeks |
| 139 | Mother Father | 2 years | Mum (-), Dad (-), 1x child (+) | 4 weeks |
| 245 | Mother | 5 years | Mum (+), Dad (-), 1x child (+) | 24 days |
| 213 | Mother Father | 3 years | Mum (+) Dad (+) Twins (+) 3yo | 3.5 weeks |
| 142 | Mother Father | 1 year | Mum (-), Dad (-), 1x child (+) | 6 weeks |
| 262 | Mother | 1.5 years | Mum (+), Dad (+) 1x child (+) | 2.5 weeks |
| 018 | Mother | 17.5 years | Mum (+), 1x child (+) 2x children not tested | 3.5 weeks |
| 143 | Mother | 4 years | Mum (-), Dad (-), 1x child (+) 1x child (-) | 6 weeks |
| 205 | Mother Father | 12–14 months | Mum (-), Dad (-), 1x child (+) 1x child (+) | 7 weeks |

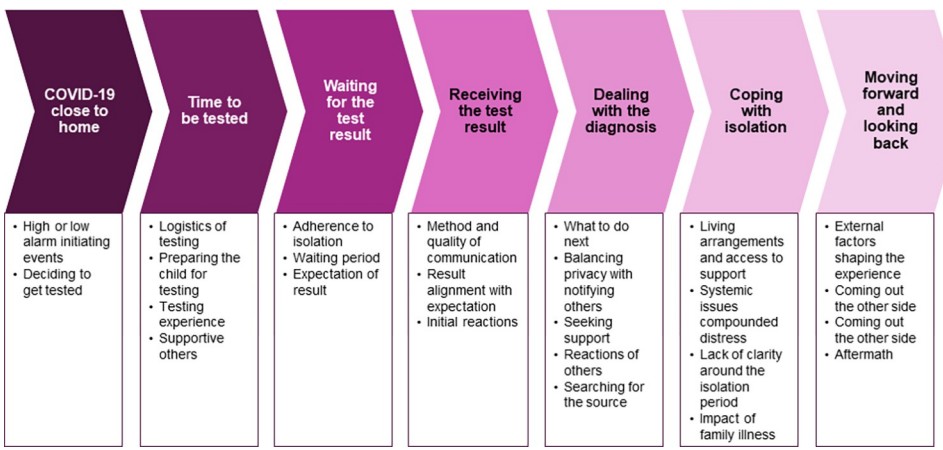

**Fig 1. The COVID-positive child journey framework–key timepoints in families' lived experience.**

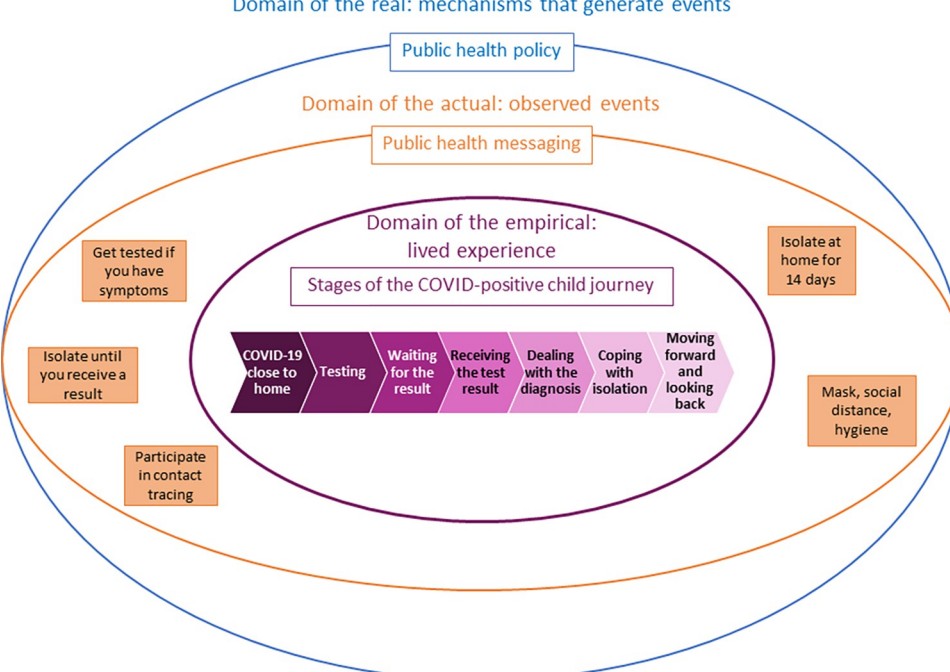

**Fig 2. Critical realist matrix illustrating relationship between public health policy and messaging and the lived experience of families with a COVID-positive child.**

[child] and I tested straightaway." (Participant 0261). Others were less motivated to get tested because a child with a runny nose was so common: "We didn't want to get him tested, we just thought it was another sickness from childcare." (Participant 142). Some participants were told to get tested by healthcare professionals or the Department of Health and Human Services (DHHS).

**Time to be tested.** The logistics of testing included selection of which site to attend. Participants mentioned previous testing sites other than RCH, including other hospital sites, GPs, and drive-through facilities. For some participants, multiple sites were attended on the same testing day; one for adult testing, and then subsequently to RCH for child testing.

> "Something that we didn't know that, as a parent, we could have got tested at the Children's [hospital] too, so we were a bit. . . We didn't know that, so we probably would have done that, if we did know." (Participant 139)

Participants spoke about preparing the child for testing. For some, this included discussions, and/or having the parent get tested first. For example, testing was described as a "big toothbrush that's going to go up the back of Mummy's throat and then it's going to go up her nose to try and get boogas [mucous]. We count how many boogas came out" (Participant 0214) or a "big tickle up your nose . . . [that afterwards felt like] sniffing pepper" (Participant 204) and sometimes a video of a child being tested was shown (Participant 245).

The testing experience varied for children. Some children were described as accepting of the test ("He didn't love it, but he didn't have any issue with going." Participant 205) while others were nervous or scared of the test; some even refused to do the test.

> "She wouldn't get it done. She would just run to the other room and would get almost like hyperventilating." (Participant 139)

Supportive others–or lack thereof–played an important role in the testing experience, including the parent and test provider.

> "This doctor was very formal about 'sit on your lap, restrain her forehead, restrain her chin' and I think that freaked [child out]." (Participant 286)

> "[The nurse] seemed really nice. She talked to [child], she was quite upbeat." (Participant 139)

Parent's role in testing was important, including preparing the child for the test, demonstrating having the test done and support through the test (e.g., holding child appropriately so test can be conducted/positioning for comfort).

**Waiting for the result–"we'll wait for our negative result" (Participant 094).** Most participants noted their adherence to the isolation requirement and immediately returned home to await the result.

> "We went into isolation in a three-by-three bedroom with an ensuite, just me and him, for our 14 days of isolation." (Participant 245)

However, one noted:

> "I was actually sure that he [child] didn't have it and I was sure that I didn't have it either, and so we drove to K-Mart, and we got him a toy and then we went on a bike ride." (Participant 211)

The waiting period for test results to be received varied, even within family members tested at the same time, from less than 24 hours to multiple days. This variation within families caused some confusion. After receiving negative test results for other members of the family, while waiting for the child's result, one participant said "I honestly thought I could leave. I thought I could go to work." (Participant 142).

Many participants viewed getting tested as just precautionary, and their expectation was that the child's test result would be negative. "Oh, we'll go off and get our COVID-19 test and we'll wait for our negative result." (Participant 094). For some, their expectation was influenced by receiving the results for other family members first.

> [Parent had tested positive] "I basically knew, he'd [child] been isolating with me, I knew it was going to come back positive, I just knew." (Participant 245)

> [Parent had tested negative] "I was okay, cool. Obviously [child] is negative as well. Then the day after, we got a call from a doctor from RCH and my head wasn't even there. [child tested positive] (Participant 211)

**Receiving the test result–"boom, he's COVID-19 positive" (Participant 245).** The method and quality of communication of test results varied. Negative results (for other members of the family) were delivered by text. Participants received positive results by phone, from either a doctor or someone from the hospital, with one thinking it may have been from the Department of Health (Participant 286). Some participants noted that there was very little additional information provided, while others indicated that the doctor had provided information and a contact phone number for any queries.

"They called and it was very vague. They really didn't know. I felt like we were the guinea pigs in the scenario. . . . [We were told] Just to stay put. 'Don't go out and we'll get in touch with you'." (Participant 142)

"[The doctor from RCH] said, 'Any question about anything call me.' I just felt quite supported because you know I guess one of the questions when you have a diagnosis is what does this mean? What happens now? They were very keen to make sure that we knew anytime, day or night, if you have a question call this number." (Participant 261)

For the most part, test results did not align with the expectation; that is, for most participants, receiving a positive result was unexpected, regardless of whether their child had strong symptoms or not.

"We got the phone call, and the lady was like 'She's positive'. And we were like 'What?'" (Participant 286; child feverish, sick, sleeping)

"I burst into tears because I wasn't expecting it. I was only sort of getting tested to tick a box to say, '[child] is fine to go to kindy'." (Participant 211)

Participants reported experiencing a range of initial reactions upon receipt of the result: shock, disbelief, embarrassment, fear, panic, guilt. One noted that they "felt like a leper, so ashamed" (Participant 142) and another felt "contaminated" (Participant 262). While one participant noted "I didn't freak out as much as I thought I would, but it wasn't a great feeling" (Participant 261), another said "the first reaction I had when I heard that [child] had COVID-19 was like 'Who's going to die?'" (Participant 211).

**Dealing with the diagnosis "What are we meant to. . .?" (Participant 204).**   During this stage of the 'testing positive journey', participants noted substantial confusion and concerns. For the most part, participants spoke about how they queried what to do next. Many noted that detailed instructions were not provided.

"No-one really was directing us, and I don't think anyone knew to be honest." (Participant 142)

"There were a couple of days of sort of a bit of radio silence and us being a bit like ah, what are we doing?" (Participant 204)

Participants described balancing privacy with notifying others about their child's positive result. Many decided to advise others of the test result and recommend that they get tested, while others felt it was important to maintain their family's privacy. Some participants indicated that they immediately advised the childcare, school, or work, as relevant. Education facilities were responsive and supportive, with many facilities closing and cleaning prior to receiving official contact from the government.

"Because the DHHS were way slower to contact anybody, including the childcare, and to shut that down" (Participant 139)

Some participants were concerned about being identified publicly as employers and schools of positive cases were often reported in the media.

Beyond notifying organisations, some participants were open with friends, family, neighbours, and colleagues about the fact that there was a positive case in their home, seeking support: "People can't support you if they don't know what's wrong" (Participant 143). Support

from family, friends, neighbours, and education facilities were reported, ranging from food parcels dropped off, toys and school activities for children, offers to walk the family dog, etcetera. One specifically noted "I didn't realise it was something to be kept secret. I didn't realise there would be a stigma" (Participant 204).

Others refrained from sharing the diagnosis with others, concerned about stigma: "I'm just scared they would treat me differently, like not want to come near me" (Participant 245). Speaking about it with others (or not) included decisions around wider family, friends, neighbours, and others in their community. This variation was also reported with the children testing positive; some families told the child, while others did not. Subsequently, some children spoke about it with others, and some did not. One school-aged child had a particularly difficult time with his peers.

> "They [student peers] posted this message with all these ew alien emojis things, saying guys, guys, [child] is positive, blah, and all these yucky emojis. . . . [child] was absolutely devastated, of course. He is already locked down in his room on his own and we already have been through all this testing and stuff which was okay but sort of really full on and scary." (Participant 204)

Many noted that they were the first in their circle to have COVID-19. The reactions of others ranged from anxious for the family and/or worried for themselves as potential contacts with some asking what they should do.

> "It's funny when you tell people, understandably their immediate concern is for themselves, because of all of the concern around COVID-19. So, ah, yeah I think most of the time we told people and their immediate reactions is 'When did we see you last? Are we at risk?' . . . I think it's interesting because it's not like a normal illness that you might tell someone, and people would be quite sympathetic." (094).

While some participants could clearly trace how COVID-19 had entered their household, others described searching for the source of the infection. Some described ongoing frustration and uncertainty, wondering if the test result had been a false positive and the implications of this, while others resigned themselves to the mystery: "My theory is either playgrounds or school and I don't think we'll ever know." (Participant 211).

**Coping with isolation period "I think uncertainty is one of the worst things that happened." (Participant 139).** Participants noted how the living arrangements and access to support helped or hindered isolating at home; some were in apartments, while others had multiple living spaces and access to the outdoors. For some, the logistics of day-to-day living were difficult, with limited support available.

> "It was very stressful just trying to get food in the house" (Participant 211)

Many parents also had to juggle work, with workplace responses increasing stress with requirements to work while unwell (e.g., work culture, limited sick leave) and others in the workplace testing positive (e.g., colleagues, aged care residents).

> "I'm not sure of the situation as to really why and what happened but I felt like I was being accused, well no, maybe I just felt guilty because of the whole situation" (Participant 245)

Systemic issues compounded distress. For many participants, the contact tracing process was particularly inefficient, inaccurate, and difficult; (e.g., manual system, not all contacts provided by participants were contacted by DHSS, family members were getting contacted multiple times on the same day, records were inaccurate). Some participants were contacted days

after receiving the diagnosis or by interstate contact tracers and were unsure what to do in the interim.

> "Then they didn't get in contact with us. I don't know how long it took, maybe two or three days or something. The biggest stress for me was the DHHS experience." (Participant 211)

> "We were getting these phone calls telling us the wrong information. Because they were treating us as close contacts [when we were actually positive.]" (Participant 262)

Some participants noted that the use of uniformed personnel (e.g., police, armed forces) was concerning, including bringing unwanted attention to families. They also noted that these check-ins focused on ensuring isolation compliance, rather than the family health.

> "And then suddenly I look through the gate and there were three policemen with masks, standing there. That was a bit of a surprise for extremely law-abiding citizens. . .Different police turned up another two times on the same day." (Participant 204)

Many participants expressed concern and frustration at the lack of clarity around the isolation period, receiving inadequate or conflicting advice from the health system. For many, once the initial 14-day period had concluded, the isolation period had to be extended as parents had been unable to isolate from the positive child. Many had to isolate for 4 weeks, with one family isolating for 7 weeks.

> "We were getting conflicting information between the COVID-19 1800 hotline and the actual DHHS information services." (Participant 214)

> "They were like 'Can't you be in a separate part of the house from her?' Like no, she's three." (Participant 286)

> "That was a bit of a bombshell when they told us we had to wait another two weeks. We thought once he [child] was clear, we were clear, but that was a huge bombshell when we found out we were another two weeks. We couldn't believe it." (Participant 142)

System limitations also meant that families had to stay in isolation longer than required.

> "And then we got told 'You actually meet criteria to leave but because there's such a backlog there's going to be a three-day delay'." (Participant 262)

The impact of family illness varied. In our cohort, children mostly had typically mild symptoms, but some parents reported worrying about symptoms and watching for worsening symptoms. In some families, multiple people were ill, forcing parents to juggle parenting, care-taking, and in some cases, work responsibilities. While one child enjoyed hotel quarantine and not going to school, others feared giving COVID-19 to other family members, disliked isolation and missed their friends, and wanted to go outside.

**Moving forward and looking back "We were just part of this really big thing that was happening" (Participant 262).** As they looked back, many participants reflected on the external factors shaping their experience. Most found it hard to find accurate and consistent information and felt that the communication from the government was frustrating.

> "And the DHHS side of it . . . made the whole experience just so much more stressful than it needed to be." (Participant 262)

Some participants were understanding towards the individuals working for the public health response, particularly those involved in contact tracing, while others were concerned about wide-spread limitations and under-resourcing, which impacted their future trust in government and the health system.

> "It's a bit disconcerting. I mean, they did a great job while were in lockdown and everyone was really fantastic, but unfortunately, no-one knew too much about it and it was early stages. So, it was frustrating, but you couldn't get frustrated with anyone for doing their job." (Participant 142)

> "So at the time we had COVID, it was at the peak of the wave. I think the DHHS system had essentially collapsed at that point. There were some huge errors. I don't want to rant about the government but there were huge errors in terms of the nursing homes and testing and stuff. Anyway, they were just in a complete shamble." (Participant 262)

Participants described coming out the other side after their family's COVID-19 experience. Some took a bigger picture perspective, looking ahead with optimism and putting their experience into a broader context.

> "I don't feel as scared for us." (Participant 230)

> "I also feel like it's going to be around for so long that you can't stress too much about it." (Participant 139)

> "We were just part of this really big thing that was happening rather than us being like the big thing. If that makes sense." (Participant 262)

Others were still grappling with the experience. Two participants compared it to living through a war, with one saying it felt like "A kind of a war situation and we just had to do what we had to do" (Participant 204). Another participant's comments indicated that the experience was still affecting them. They described telling themselves they had to "let it go" while making a concerted effort to focus on the future: "We just have to keep positive and keep the good things, you know, these are the good things that are going to happen to us, keep that going" (Participant 245). Another was concerned about future consequences of COVID infection: "I wonder what the repercussions of this is going to be?" (Participant 94).

The aftermath of the whole process, from testing to diagnosis to isolation, included impacts on health behaviours as well as social, physical, and mental health consequences. A few participants indicated that having symptoms alone would not mean that they would get tested in future, either because they were wary of going through the isolation experience again or because their child was traumatised by the testing process.

> "She's [child] adamant that no doctor's going to go anywhere near her nose ever again." (Participant 213).

> "But my five-year-old, she's, I would say, this is strong language but, verging on permanently traumatised [laughs], well not permanently, but whenever we drove past the Children's, she could almost start crying and now she's terrified to go to any doctor and even have medicine." (Participant 204)

Others described mental health impacts for their children and themselves.

"He's [child] become really, really controlling of his food and really, really health conscious, but to a point that I actually think he's a bit unhealthy. He's lost weight. . .As you could sort of understand from the way he handled being positive and he's so self-managing and self-controlling, in a way." (Participant 204)

"Afterwards my anxiety was like a thousand percent. I didn't step foot in a shop for probably six weeks." (Participant 261)

## Discussion

Our results present an exploration of the journey families undergo when having a child test positive for COVID-19, which can inform current and future pandemic planning, as well as future models of care. While the families we interviewed varied in a number of ways, they all described experiences that moved sequentially through the same seven key stages, which we outline in our journey framework: COVID-19 close to home, Testing, Waiting for the result, Receiving the test result, Dealing with the diagnosis, Coping with isolation, Moving forward and looking back. Within these timepoints, we have identified a range of shared and discrete experiences for our families, influenced by both individual factors and systemic processes. To our knowledge, this is the first detailed depiction of families' experiences of a child testing positive in Australia.

Importantly, the critical realist approach we adopted for this study has supported identifying examples which illustrate differences or disconnections between the reality of government policies and public health messaging (i.e., domain of the real and actual) [9] and the reality of the lived experience (i.e., domain of the empirical) [9]. For example, messaging to get tested was insufficient as children under 5 years could not be tested at all testing centres, meaning families attended multiple centres so parent/s and child/ren could be tested. Messaging to isolate apart from the COVID-positive individual was not possible for parents, and the stated isolation period of 14 days was not accurate for many families. As such, we have identified areas of concern and present recommendations for strategies that align with the journey stages to address the unmet needs for families (Table 2).

Systemic processes proved difficult in many of the stages: the Testing (e.g., can adults and children be tested at all testing sites, where to go to have a child tested not clear), Waiting for test results (e.g., family members' test results did not all come together), Receiving the test result (e.g., insufficient information provided about what to do next, inadequate support for those who were not expecting the result), Dealing with Diagnosis (e.g., not knowing what to do next, waiting to be contacted), Coping with Isolation (e.g., inaccurate and repetitive contact tracing requirements) and Moving Forward and looking back (e.g., inadequate mental health support, systemic challenges exacerbating stress). For some, the difficult experiences with systems resulted in reduced trust in government agencies as well as caution around future healthcare relationships.

For many of our participants, despite reporting only mild physical symptoms, the mental health impacts for the child and family were profound and extended after they were symptom-free. Family members experienced varying levels of concern and worry, confusion and frustration, guilt, and shame. The 14-day isolation period advertised through government messaging was not accurate for most participants, as other COVID-negative family members had to isolate for an additional 14-day period once the child completed the isolation period and/or was symptom-free. The lack of clarity in the messaging around the isolation period for families was a key source for subsequent emotional and practical difficulties; addressing this would support realistic management of expectations.

**Table 2. Recommended strategies to address gaps between public health messaging and lived experience at journey framework timepoints.**

| PUBLIC HEALTH MESSAGING AT KEY TIMEPOINTS | Get tested if you have any symptoms | Here's where to go to get tested | Go straight home and isolate until you receive your result | What to do if you have COVID | Participate in contact tracing to identify close contacts | Isolate at home for 14 days | General public health messaging |
|---|---|---|---|---|---|---|---|
| *Example messages* | "Don't take this disease lightly. If you feel unwell with any symptoms of coronavirus, however mild, you should stay home and get tested." [15] | "There is no excuse for not getting tested. We have people knocking on your door, coming to your neighbourhood–we are bringing the testing to you. There are also several drive-through and fixed sites where people can go." [15] | "If you have any fever, chills, cough, sore throat, shortness of breath, runny nose, and loss of sense of smell or taste—stay home, don't go in to work and don't visit friends and family. Get tested and stay at home until you get the result." [16] | "If you have tested positive for coronavirus (COVID-19): You must isolate yourself until the Department of Health and Human Services tells you it is safe. It is important that you follow this guidance–as required by law." [17] | "After you have been told about your positive test result, you must inform your employer and you can inform your close contacts." [17] | "Quarantine means you cannot leave your home or accommodation for any reason, except for medical care or in an emergency." [18] | "Stay 1.5 metres away from anyone you don't live with and avoid crowds, especially indoors. If you can keep working from home–you must keep working from home." [15] "A face covering will be mandatory whenever you leave home –wherever you live." [19] |
| LIVED EXPERIENCE AT KEY TIMEPOINTS | COVID-19 may be at home | Time to be tested | Waiting for the test result | Receiving the result | Dealing with the diagnosis | Coping with isolation | Moving forward and looking back |
| *Recommended strategies to improve experience* | | • Train and deploy paediatric testers experienced in child-specific requirements to more testing sites. • Publicise sites with testing capability for children under 5 and ensure these sites can also test family members • Prepare children with educational resources like videos | • Make explicit instructions to return directly home following testing, including why important to do so • Ensure consistency in training and messaging of Government staff, hotline operators and testing facilitators to minimise confusion | • Acknowledge emotional experience of receiving a positive result and consider information timing and format, e.g., provide key information in written form to allow recipient to digest over time • Ensure single point of contact and provide name and number to contact for queries (e.g., case manager) • Provide step-by-step details as to what happens next (immediate, short- and longer term), what to do | • Schedule post-diagnosis consultation incorporating tailored education, information and support • Maintain privacy when reporting on cases or outbreaks • Reduce stigma with media and public health messaging from a 'care and support' perspective, rather than 'shame and fear' | • Clarifying factors influencing isolation periods at beginning, not end of child's isolation period • Provide realistic isolation guidance (e.g., do not suggest children isolate alone) | • Establish peer support networks or groups for families coping with repercussions of experience, either logistical, physical or emotional • Prepare teachers and school administrators for potential stigma and bullying issues related to diagnosis during absence or upon return to school |

Participants also described privacy concerns and fear of judgment. Conflicting messaging, insufficient knowledge about disease prognosis and the lack of effective therapeutics or a preventive vaccine drove COVID-19-related stigma around the world in the early part of the pandemic [20]. However, stigma in Victoria may have been exacerbated by the relatively low prevalence of COVID-19 in 2020. Familiarity is often associated with reduced stigma [21], but at the time of this study, very few people knew someone personally who had COVID-19. Furthermore, positive cases were subject to intensive media and public attention because they had a direct impact on the type, location and duration of public health restrictions (e.g., lockdowns of specific postcodes, school closures), in turn, impacting on others' lives. While stigma may

have decreased over the course of the pandemic, the judgment and shame experienced by study participants and other early cases may have serious and long-lasting physical and mental health consequences. Potential stigma should be addressed early in any future disease outbreaks, with public health messaging focusing on protecting the privacy of infected individuals and encouraging solidarity, community and empathy [22].

As with much of the pandemic, organisations involved were having to address issues in real-time, with refinements likely undertaken. Understanding the experience from a family perspective (rather than a population/public health perspective) can provide insights to improve systemic processes. Some changes are already underway. One example of a helpful change is that there is now post-testing support provided via personal text messaging which includes a link to details of what to do after being tested. Ideally, these details would be tailored to cover if one or more members of the same household were tested at the same time and/or if some members cannot isolate independently from others (e.g., due to age, or household restrictions). Other changes may raise concerns. For example, as families increasingly use rapid antigen tests at home, families will be receiving potentially distressing news without any expert support present. Having a consistent contact (e.g., case manager model) will provide families with continuity, allowing trust to build and provide support. Other families who have had positive cases (i.e., peer support) could provide lived experience insights and successful strategies. The importance of communicating test results accurately [23], including the importance of tailoring information [24] should be incorporated. Once results are received, ensuring linkage with a treating GP and accurate, relevant information was fundamental for these families. Previous work has illustrated that government communication can be difficult to read [25,26]. Although information may change during the pandemic [27], our previous research has indicated that the public accept this, but it must be clearly explained [28].

## Strengths, limitations and future research

This is the only qualitative study we are aware of in Australia that focuses on the experiences of families where a child tested positive for COVID-19 in the first year of the pandemic. This study has several strengths, including our use of a critical realist ground theory approach to explore the lived experience of families. This approach allowed us to define distinct time points in this experience and identify systemic limitations impacting families. Importantly, we propose this initial framework to identify where specific types of support are required to improve the lived experience of families where a child has tested positive for COVID-19.

Despite this new and important outcome, several limitations need to be considered, specific to our participant group and the context in which the study was undertaken. At the time of study recruitment, positive cases among children were relatively rare, and we were not able to make contact with all potentially eligible families to invite them to take part in an interview as some had moved or were otherwise not contactable by the time we were able to access their contact details. Due to the relatively small sample size, we present a proposed framework but were not able to develop and test a comprehensive theory with our participant group. Further limitations specific to our participants include that the children in our sample were young, due to those under age 5 being referred to testing at the RCH, and all experienced mild COVID-19 cases. The experience of families may differ with older children or those with severe symptoms, including hospitalisations [29]. Our participants were recruited from a setting where there was the potential for them to be invited to participate in multiple COVID-19 related studies, which included physical assessments, repeated biospecimen collection and surveys. This may have impacted their recall of events associated with their initial testing and diagnosis experience. Our sample pool was also relatively homogenous demographically, as we were unable to

include participants who could not speak and understand English due to resource limitations. The ways in which the journey timepoints may be experienced for those from culturally and linguistically diverse backgrounds, those with disability or underlying conditions, or low health literacy, and their specific experiences and needs, requires further research. Limitations relating to the context include the fact that restrictions changed throughout the study, meaning some families experienced isolation when the community was in lockdown, while other families isolated with COVID-normal life continuing without them. The physical aspects of testing have also changed over time (e.g., less intrusive swabbing and rapid antigen testing), as have the public health response measures and messaging. In this study, participants were experiencing public health systems that were being developed, implemented, and refined in real-time. Systemic changes are likely to have occurred and include improvements such as electronic contact tracing (not paper-based) as well as concerning changes such as receiving positive results via text. How these developments influence family experience warrants further exploration.

While we did not achieve saturation [30] with our pragmatic sample, we propose that this does not affect the generalisability of the journey framework as such, but may impact the content at each stage. Next steps of formulating and exploring theoretical relationships (abductive and retroductive analytic approaches) [9,31] between experiences in different stages awaits further research with a larger sample. For example, whether a high or low alarm event impacts the expectation of a positive or negative result, and how this may impact the experience of receiving a positive result should be explored. Consideration of other stakeholders, such as testers, contact tracers and care/education representatives may also contribute meaningful insights.

## Conclusion

Having a child receive a positive COVID-19 diagnosis impacts on the immediate family and extended community. Even with no or mild physical symptoms, our participants illustrate the substantial emotional and psychological impacts, and burden that family members experienced, exacerbated by public health policies that did not adequately consider the effects for those with young children.

This study highlights the importance of family-centred messaging and policies during the ongoing COVID-19 pandemic and in future pandemics or disease outbreaks. Families have unique experiences and support needs that may not be addressed by the systems and communication directed to the general public. Further research and collaboration are proposed to extend the current framework to a comprehensive theory for the experience of testing positive for COVID-19.

## Acknowledgments

The authors would like to thank the members of the COVID Wellbeing Working Group, who provided support with study design and recruitment: Dr Shidan Tosif, Dr Natasha Brusco, Professor Jennifer Watts, Dr Meredith O'Connor, Dr Danielle Wurzel, Professor Dave Burgner and Professor Craig Olsson.

## Author Contributions

**Conceptualization:** Jessica Kaufman, Tria Williams, Margie Danchin.

**Data curation:** Jessica Kaufman.

**Formal analysis:** Jessica Kaufman, Kathleen L. Bagot, Tria Williams.

**Funding acquisition:** Jessica Kaufman, Margie Danchin.

**Investigation:** Jessica Kaufman, Tria Williams.

**Methodology:** Jessica Kaufman, Margie Danchin.

**Project administration:** Jessica Kaufman, Tria Williams, Carol Jos.

**Supervision:** Jessica Kaufman, Margie Danchin.

**Writing – original draft:** Jessica Kaufman, Kathleen L. Bagot, Tria Williams.

**Writing – review & editing:** Jessica Kaufman, Kathleen L. Bagot, Tria Williams, Carol Jos, Margie Danchin.

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
