## [Decision Letter · Decision Letter 0]

22 Nov 2022

PONE-D-22-15338Exploring the lived experience of families with a COVID-19 positive child: the journey from a critical grounded theory approachPLOS ONE

Dear Dr. Kaufman,

Thank you for submitting your manuscript to PLOS ONE. After careful consideration, we feel that it has merit but does not fully meet PLOS ONE’s publication criteria as it currently stands. Therefore, we invite you to submit a revised version of the manuscript that addresses the points raised during the review process.

ACADEMIC EDITOR:Please resubmit a revised version of the manuscript while addressing these comments:

1) I couldn't find an added value to the literature. Most of the study's findings are already known and they fail to fulfill the real world's practical needs. Finally, the take-home message of the study was not clear.

2) page 12 line 110 - text reading "that is, theoretical sampling", please add "within the sample". I suggest this be added to prevent confusion with the statement above that theoretical sampling was not possible.

3) Table 1 - in two rows "Dad" is underlined. Is there a meaning to the underlining. If so, please provide a key. If not please remove the underlining.

4) page 13 - text reading "Some participants were concerned about being identified in the media, since employers and schools of positive cases were sometimes reported" - do you mean "sometimes notified"? I assume so given the context, or do you mean that employers and schools were reported to public health authorities, or reported in the media? Please clarify.

5) page 21 line 21 - Do you think that the feelings of stigma would have been grounded in the low incidence of COVID-19 within Australia at that time (particularly relative to other countries)? Do you think that there would be less stigma now that COVID-19 is more prevalent within Australia? A discussion about the stigma of low-prevalence diseases and acceptance of high-prevalence diseases might make a valuable contribution, particularly in the context of the stigma associated with the new monkey-pox 'epidemic'.

6) page 23 line 25 - word missing "due those under age 5 being" should read "due to those ..."

Please submit your revised manuscript 06 January 2023. If you will need more time than this to complete your revisions, please reply to this message or contact the journal office at plosone@plos.org. Please include the following items when submitting your revised manuscript:A rebuttal letter that responds to each point raised by the academic editor and reviewer(s). You should upload this letter as a separate file labeled 'Response to Reviewers'.A marked-up copy of your manuscript that highlights changes made to the original version. You should upload this as a separate file labeled 'Revised Manuscript with Track Changes'.An unmarked version of your revised paper without tracked changes. You should upload this as a separate file labeled 'Manuscript'.

We look forward to receiving your revised manuscript.

Kind regards,

Mohsen Abbasi-Kangevari

Academic Editor

PLOS ONE

a) If there are ethical or legal restrictions on sharing a de-identified data set, please explain them in detail (e.g., data contain potentially sensitive information, data are owned by a third-party organization, etc.) and who has imposed them (e.g., an ethics committee). Please also provide contact information for a data access committee, ethics committee, or other institutional body to which data requests may be sent

**Comments to the Author**

1. Is the manuscript technically sound, and do the data support the conclusions?

Reviewer #1: Yes

Reviewer #2: Yes

Reviewer #3: Yes

2. Has the statistical analysis been performed appropriately and rigorously? 

Reviewer #1: N/A

Reviewer #2: N/A

Reviewer #3: Yes

3. Have the authors made all data underlying the findings in their manuscript fully available?

Reviewer #1: Yes

Reviewer #2: Yes

Reviewer #3: No

4. Is the manuscript presented in an intelligible fashion and written in standard English?

Reviewer #1: No

Reviewer #2: Yes

Reviewer #3: Yes

5. Review Comments to the Author

Reviewer #1: This is a well written and well structured article presenting the COVID-19 experience of families with young children. The critical realist approach is valuable, particularly in providing a grounded commentary on COVID-19 policy and messaging which will have important policy and public health practice implications.

There are a very small number of corrections needed and a suggestion to broaden one area of discussion regarding stigma:

- page 12 line 110 - text reading "that is, theoretical sampling", please add "within the sample". I suggest this be added to prevent confusion with the statement above that theoretical sampling was not possible.

- Table 1 - in two rows "Dad" is underlined. Is there a meaning to the underlining. If so, please provide a key. If not please remove the underlining.

- page 13 - text reading "Some participants were concerned about being identified in the media, since employers and schools of positive cases were sometimes reported" - do you mean "sometimes notified"? I assume so given the context, or do you mean that employers and schools were reported to public health authorities, or reported in the media? Please clarify.

- page 21 line 21 - Do you think that the feelings of stigma would have been grounded in the low incidence of COVID-19 within Australia at that time (particularly relative to other countries)? Do you think that there would be less stigma now that COVID-19 is more prevalent within Australia? A discussion about the stigma of low-prevalence diseases and acceptance of high-prevalence diseases might make a valuable contribution, particularly in the context of the stigma associated with the new monkey-pox 'epidemic'.

- page 23 line 25 - word missing "due those under age 5 being" should read "due to those ..."

Overall, congratulations to your research team for a high quality, well conceived and conducted, and well written paper.

Reviewer #2: The authors intended to investigate the experiences within a family with a child who has a SARS-CoV-2 PCR positive test result. Even though the manuscript is well written, I couldn't find an added value to the literature. Most of the study's findings are already known and they fail to fulfill the real world's practical needs. Finally, the take-home message of the study was not clear.

Reviewer #3: Dear Authors,

Thank you for giving me the opportunity to review your article about the experience of families with a COVID-19 positive child. The topic was interesting for me. The grounded theory as the methodology of study was appropriate. All parts of article are written with enough details. The authors describe the steps of study completely. Furthermore, the discussion and conclusion of article are relevant and make sense. however, there are some important issues regarding the sampling which are mentioned in limitations part by details. In conclusion, due to the authors great explanation, I think this manuscript is suitable for publication.

6. PLOS authors have the option to publish the peer review history of their article (what does this mean?). If published, this will include your full peer review and any attached files.

Reviewer #1: **Yes: **Distinguished Professor Lynn Kemp

Reviewer #2: **Yes: **Mohammadreza Azangou-Khyavy

Reviewer #3: **Yes: **Ali Golestani

---

## [Author Response · Author response to Decision Letter 0]

7 Dec 2022

We thank the reviewers for their thoughtful feedback and outline our responses to their comments below. 

1) The authors intended to investigate the experiences within a family with a child who has a SARS-CoV-2 PCR positive test result. Even though the manuscript is well written, I couldn't find an added value to the literature. Most of the study's findings are already known and they fail to fulfill the real world's practical needs. Finally, the take-home message of the study was not clear.

Author response: This study is the first to our knowledge to identify the disconnect between the lived experiences of families with young children and early pandemic government messaging and policies designed for the general public. As our participants expressed, inadequate consideration of children and families in public health messaging and policy contributed to psychological distress and stigmatisation. The critical realist analysis approach, in particular, is a unique feature of this study. Using this lens, we not only identified unmet needs of families at specific timepoints, we also developed specific recommended strategies to improve their experience and inform future pandemic planning (Table 2). 

We have highlighted the take-home message of the manuscript in the abstract and include the following text in the conclusion (p24): 

“This study highlights the importance of family-centred messaging and policies during the ongoing COVID-19 pandemic and in future pandemics or disease outbreaks. Families have unique experiences and support needs that may not be addressed by the systems and communication directed to the general public.”

2) page 12 line 110 - text reading "that is, theoretical sampling", please add "within the sample". I suggest this be added to prevent confusion with the statement above that theoretical sampling was not possible.

Author response: We agree that this may be confusing so have amended the text in line 115 to remove the second reference to theoretical sampling.

3) Table 1 - in two rows "Dad" is underlined. Is there a meaning to the underlining. If so, please provide a key. If not please remove the underlining.

Author response: The underlining was in error and has been removed. 

4) page 13 - text reading "Some participants were concerned about being identified in the media, since employers and schools of positive cases were sometimes reported" - do you mean "sometimes notified"? I assume so given the context, or do you mean that employers and schools were reported to public health authorities, or reported in the media? Please clarify.

Author response: We have clarified that parents and children who tested positive, particularly if they were the index case, were concerned about being identified publicly. The media reported details about the location of positive cases identified in schools or workplaces and the cases were often able to be identified in small outbreaks, causing distress for families. The amended sentence reads “Some participants were concerned about being identified publicly as employers and schools of positive cases were often reported in the media.” (lines 282-283, p 13)

5) page 21 line 21 - Do you think that the feelings of stigma would have been grounded in the low incidence of COVID-19 within Australia at that time (particularly relative to other countries)? Do you think that there would be less stigma now that COVID-19 is more prevalent within Australia? A discussion about the stigma of low-prevalence diseases and acceptance of high-prevalence diseases might make a valuable contribution, particularly in the context of the stigma associated with the new monkey-pox 'epidemic'.

Author response: We appreciate the reviewer’s thoughtful suggestion. We agree the issue around stigma for COVID-19 positive cases, especially index children who triggered a school closure, was intense at the start of the pandemic and of great concern to many parents. This has not been well reported in the literature, especially from Australia. 

COVID-19 related stigma was reported around the world in the early part of the pandemic, suggesting that it was driven by confusion, fear and lack of therapeutics or vaccines at that time, more than by familiarity with the disease. However, it is possible that the limited case numbers in Australia and the intense restrictions associated with individual cases and their movements (e.g. postcode-wide lockdowns, school closures) exacerbated stigma. We have expanded our discussion and included additional literature related to the issue of stigma (lines 456-468, pp21-22).

6) page 23 line 25 - word missing "due those under age 5 being" should read "due to those ..."

Author response: This has been amended. 

JOURNAL REQUIREMENTS

Author response: We have amended the manuscript and file names as requested. 

a) If there are ethical or legal restrictions on sharing a de-identified data set, please explain them in detail (e.g., data contain potentially sensitive information, data are owned by a third-party organization, etc.) and who has imposed them (e.g., an ethics committee). Please also provide contact information for a data access committee, ethics committee, or other institutional body to which data requests may be sent

Author response: As per the study protocol approved by the Royal Children’s Hospital Human Research Ethics Committee, de-identified participant data will be made available on request for use by future researchers from a recognised research institution whose proposed use of the data has been ethically reviewed and approved by an independent committee. Contact details: +61 3 9345 5044 or email rch.ethics@rch.org.au

Author response: We have included the full ethics details and noted that written informed consent was obtained.

---

## [Decision Letter · Decision Letter 1]

5 Feb 2023

PONE-D-22-15338R1Exploring the lived experience of families with a COVID-19 positive child: the journey from a critical grounded theory approachPLOS ONE

Dear Dr. Kaufman,

Thank you for submitting your manuscript to PLOS ONE. After careful consideration, we feel that it has merit but does not fully meet PLOS ONE’s publication criteria as it currently stands. Therefore, we invite you to submit a revised version of the manuscript that addresses the points raised during the review process.

ACADEMIC EDITOR: We appreciate the authors' resubmission of their article. However, I have a few minor clarifications:

1. In lines 79-80: authors have revised the manuscript to include obtaining a "written informed consent". However, the authors claimed that this was done over the phone by the research assistants. Authors should explain and provide further information on how they obtained written consent.

2. In lines 95-96: How did you assure the rigor and trustworthiness of the online data collection, i.e. using Zoom? To guarantee procedural clarity, I would consider adding information or a reference to the approach you followed.

3. In lines 73-174: Readers may be perplexed by the term "boogas" in the quotations because it is a slang term. This problem could be solved by using a generally recognized phrase in brackets after the word.

4. I acknowledge that the two children's interviews were insufficient to undertake data analysis; nevertheless, given the journey component of the proposed framework, is there any possibility that one or both of the children interviewed as positive cases may be connected in the article as an illustrative case? This could be linked in the results - this would call policymakers' and academics' attention to the fact that, while your study was unable to catch more children, the stories and narratives of these children could improve the primary takeaways and criticality of the data.

We look forward to receiving your revised manuscript.

Kind regards,

Mark Donald C Reñosa, PhD

Academic Editor

PLOS ONE

Journal Requirements:

1. If the authors have adequately addressed your comments raised in a previous round of review and you feel that this manuscript is now acceptable for publication, you may indicate that here to bypass the “Comments to the Author” section, enter your conflict of interest statement in the “Confidential to Editor” section, and submit your "Accept" recommendation.

Reviewer #2: (No Response)

2. Is the manuscript technically sound, and do the data support the conclusions?

Reviewer #2: Yes

3. Has the statistical analysis been performed appropriately and rigorously? 

Reviewer #2: Yes

4. Have the authors made all data underlying the findings in their manuscript fully available?

Reviewer #2: No

5. Is the manuscript presented in an intelligible fashion and written in standard English?

Reviewer #2: Yes

6. Review Comments to the Author

Reviewer #2: (No Response)

7. PLOS authors have the option to publish the peer review history of their article (what does this mean?). If published, this will include your full peer review and any attached files.

Reviewer #2: No

---

## [Author Response · Author response to Decision Letter 1]

15 Feb 2023

We thank the editor for their feedback on this manuscript. Our responses are outlined below. 

1) In lines 79-80: authors have revised the manuscript to include obtaining a "written informed consent". However, the authors claimed that this was done over the phone by the research assistants. Authors should explain and provide further information on how they obtained written consent.

Author response: Using the REDCap eConsent functionality, we were able to collect written informed consent electronically (i.e., participants could provide a digital signature to the consent form). We have added clarification around this point in lines 80-81.

2) In lines 95-96: How did you assure the rigor and trustworthiness of the online data collection, i.e. using Zoom? To guarantee procedural clarity, I would consider adding information or a reference to the approach you followed.

Author response: We appreciate this suggestion. Our interview procedures were designed to address the main potential challenges of Zoom interviewing (e.g. lack of rapport, unstable internet issues, privacy). We have described this briefly and included a reference that informed our approach. 

3) In lines 73-174: Readers may be perplexed by the term "boogas" in the quotations because it is a slang term. This problem could be solved by using a generally recognized phrase in brackets after the word.

Author response: Thank you for identifying this issue. We have defined the term as ‘mucous’ in the quote (line 175).

4) I acknowledge that the two children's interviews were insufficient to undertake data analysis; nevertheless, given the journey component of the proposed framework, is there any possibility that one or both of the children interviewed as positive cases may be connected in the article as an illustrative case? This could be linked in the results - this would call policymakers' and academics' attention to the fact that, while your study was unable to catch more children, the stories and narratives of these children could improve the primary takeaways and criticality of the data.

Author response: We appreciate this suggestion, but unfortunately the two children mentioned did not complete full interviews. They were both young and offered only a handful of short responses or elaborations when prompted by their parents. We did attempt to recruit families with older children who could participate more fully in the interviews, but unfortunately we were not successful with this effort. Future research focusing specifically on the views of children and adolescents would be very valuable, given the reports of mental health challenges experienced by children during this period.

---

## [Editor Report · Decision Letter 2]

16 Feb 2023

Exploring the lived experience of families with a COVID-19 positive child: the journey from a critical grounded theory approach

PONE-D-22-15338R2

Dear Dr. Kaufman,

We’re pleased to inform you that your manuscript has been judged scientifically suitable for publication and will be formally accepted for publication once it meets all outstanding technical requirements.

Kind regards,

Mark Donald C Reñosa, PhD

Academic Editor

PLOS ONE

---

## [Editor Report · Acceptance letter]

20 Feb 2023

PONE-D-22-15338R2 

Exploring the lived experience of families with a COVID-19 positive child: the journey from a critical grounded theory approach 

Dear Dr. Kaufman:

I'm pleased to inform you that your manuscript has been deemed suitable for publication in PLOS ONE. Congratulations! Your manuscript is now with our production department. 

Kind regards, 

on behalf of

Dr. Mark Donald C Reñosa 

Academic Editor

PLOS ONE